# Exploring the Surface: Sampling of Potential Skin Cancer Biomarkers Kynurenine and Tryptophan, Studied on 3D Melanocyte and Melanoma Models

**DOI:** 10.3390/biom14070815

**Published:** 2024-07-09

**Authors:** Sylwia Hasterok, Skaidre Jankovskaja, Ruzica Miletic Dahlström, Zdenka Prgomet, Lars Ohlsson, Sebastian Björklund, Anna Gustafsson

**Affiliations:** 1Department of Biomedical Science, Faculty of Health and Society, Malmö University, 205 06 Malmo, Sweden; skaidre.jankovskaja@mau.se (S.J.); ruzica.miletic@outlook.com (R.M.D.); zdenka.prgomet@mau.se (Z.P.); lars.ohlsson@mau.se (L.O.); sebastian.bjorklund@mau.se (S.B.); 2Biofilms Research Center for Biointerfaces, Malmö University, 205 06 Malmo, Sweden; 3Section for Oral Biology and Pathology, Faculty of Odontology, Malmö University, 214 21 Malmo, Sweden

**Keywords:** kynurenine, *IDO-1*, tryptophan, skin cancer biomarkers, melanoma, full-thickness 3D skin models, IFN-γ, UVB, non-invasive sampling

## Abstract

Early detection of cancer via biomarkers is vital for improving patient survival rates. In the case of skin cancers, low-molecular-weight biomarkers can penetrate the skin barrier, enabling non-invasive sampling at an early stage. This study focuses on detecting tryptophan (Trp) and kynurenine (Kyn) on the surface of reconstructed 3D melanoma and melanocyte models. This is examined in connection with *IDO-1* and *IL-6* expression in response to IFN-γ or UVB stimulation, both crucial factors of the melanoma tumor microenvironment (TME). Using a polystyrene scaffold, full-thickness human skin equivalents containing fibroblasts, keratinocytes, and melanocytes or melanoma cells were developed. The samples were stimulated with IFN-γ or UVB, and Trp and Kyn secretion was measured using HPLC-PDA and HPLC-MS. The expression of *IDO-1* and *IL-6* was measured using RT-qPCR. Increased Trp catabolism to Kyn was observed in IFN-γ-stimulated melanoma and melanocyte models, along with higher *IDO-1* expression. UVB exposure led to significant changes in Kyn levels but only in the melanoma model. This study demonstrates the potential of skin surface Trp and Kyn monitoring to capture TME metabolic changes. It also lays the groundwork for future in vivo studies, aiding in understanding and monitoring skin cancer progression.

## 1. Introduction

As of today, the gold standard for diagnosing early-stage skin cancer relies on visual examination and biopsy [1]. Still, research suggests that this approach has its limitations, with specificity falling below 30% and sensitivity hovering at around 84% [1,2]. Consequently, there is a pressing need for additional tools, preferably non-invasive, to enhance the accuracy of diagnosis [2], and readily available, early-onset cancer biomarkers could significantly improve patient survival. In skin cancer research, the identification of low-molecular-weight (LMW) biomarkers is highly promising. These smaller molecules can easily cross the skin barrier, allowing for non-invasive topical sampling methods [3,4].

Tryptophan (Trp) is an essential amino acid [5], involved in several major metabolic pathways, including protein synthesis, serotonin production, and the conversion of Trp into tryptamine through decarboxylation [6,7]. Another of these metabolic pathways is the kynurenine pathway (KP), in which Trp is enzymatically converted into its metabolite, kynurenine (Kyn). Kyn is subsequently transformed into various biologically active metabolites, including kynurenic acid, picolinic acid, and NAD+, through quinolinic acid or anthranilic acid intermediaries. Undeniably, the KP is immensely significant in human metabolism since approximately 95% of dietary Trp undergoes a metabolic transformation through this pathway [6,7]. The conversion of Trp into Kyn is catalyzed by three rate-limiting enzymes, i.e., indoleamine 2,3-dioxygenase 1 and 2 (*IDO-1* and IDO-2), and tryptophan 2,3-dioxygenase (TDO) [8]. Among these enzymes, *IDO-1* is extensively upregulated in several cancer types, including melanoma [9]. The range of melanomas featuring upregulated *IDO-1* expression spans from approximately 15–92%, with ongoing research aimed at refining these statistics [10,11,12,13]. Nevertheless, it is established that the overexpression of *IDO-1* and the decreased Trp/Kyn ratio exert immunosuppressive effects, resulting in significantly lower patient survival rates [14,15].

The tumor microenvironment (TME) constitutes a complex molecular network that facilitates interactions between cancer cells and neighboring cells, including healthy cells and immune cells. This intricate interplay is orchestrated by a variety of signaling molecules such as cytokines, chemokines, growth factors, and various enzymes. These pro-inflammatory molecules have been associated with the emergence of chronic inflammation, which is one of the hallmarks of cancer development [16]. Consequently, chronic inflammation can lead to systematic immunosuppression, providing a favorable environment for cancer initiation and progression [17]. Moreover, the interaction between melanoma cells and the TME has been shown to result in increased production of interleukin 6 (*IL-6*), interleukin 8 (IL-8), interleukin 10 (IL-10), tumor necrosis factor alpha (TNF-α), vascular endothelial growth factor (VEGF), interferon gamma (IFN-γ), and many more [18]. Among these molecules, IFN-γ is a well-studied cytokine that plays a pivotal role in initiating both innate and adaptive immune responses. While it is widely recognized for its anti-tumor properties, it also possesses the capacity to promote tumor growth by facilitating immune evasion through immunosuppression [19]. IFN-γ is mainly produced by tumor-infiltrating lymphocytes, which may lead to the upregulation of *IDO-1* through a negative feedback mechanism. This IFN-γ-mediated signal transduction process involves the phosphorylation of signal transducer and activator of transcription 1 (STAT-1) tyrosine, which triggers its dimerization and binding to the gamma interferon activation site (GAS) sequence within the *IDO-1* gene. Secondly, the signal initiates the synthesis of IFN-γ regulated factor 1 (IRF1) by nuclear factor-κB (NF-κB) and STAT-1, enabling IRF1 to bind to interferon-sensitive response element (ISRE) sequences within the *IDO-1* gene [20]. Collectively, these mechanisms coordinate the maximal induction of *IDO-1* transcription mediated by IFN-γ.

Immunosuppression arises when the overexpression of *IDO-1* in the skin depletes Trp and generates excessive amounts of Kyn, which decreases the Trp/Kyn ratio in the melanoma TME [9]. More specifically, the exhaustion of Trp triggers the activation of general control nonderepressible 2 kinase (GCN2) in T cells, consequently hindering T cell proliferation and initiating the differentiation of regulatory T cells (Treg) [9,21]. Furthermore, the binding of Kyn to the aryl hydrocarbon receptor (AhR) also launches Treg differentiation and contributes to the formation of an immunosuppressive phenotype mediated by macrophages and dendritic cells [21]. Therefore, the transformation of Trp through the KP, driven by the overexpression of *IDO-1* in tumor cells, as well as neighboring cells (i.e., fibroblasts and infiltrating immune cells), likely constitutes a significant component of the mechanisms responsible for immune evasion. Moreover, given that the shortage of essential nutrients in the TME is another hallmark of cancer, driven by the heightened consumption of glucose, glutamine, Trp, and arginine [21], such an explanation seems very plausible. This, in turn, ultimately facilitates melanoma progression. In addition to elevated Kyn concentrations, the TME is also characterized by conditions like hypoxia and elevated levels of immunosuppressive metabolites such as lactate, reactive oxygen species (ROS), and polyamines. These conditions serve to not only support tumor metabolism and growth but also stifle anti-tumor immune responses [21,22,23,24].

However, cells exhibiting heightened *IDO-1* expression appear to also trigger an autocrine positive feedback loop. This loop is initiated through the activation of AhR by Kyn and involves another cytokine with pleiotropic effects, *IL-6* [25,26]. Specifically, the activation of AhR triggers the up-regulation of *IL-6*, which subsequently mediates signal transducer and activator of transcription 3 (STAT-3) signaling, promoting the expression of *IDO-1* in tumor cells [26]. *IL-6* is also a well-known participant in chronic inflammation, a prevalent characteristic of the TME [27]. Additionally, the proinflammatory cytokine *IL-6* is released in the skin following exposure to ultraviolet B (UVB, 280–320 nm) radiation, which is a major factor responsible for the initiation of melanoma development [28,29,30,31]. While both indirect DNA damage (caused by ROS) and direct DNA damage (due to UVB exposure) are required to initiate a malignant transformation, it is the mechanisms underlying melanoma’s immunosuppressive effects that primarily drive disease progression [32].

Our comprehension of malignant melanoma has significantly advanced in recent years. Numerous studies have shed light on the initiation and progression of melanoma (reviewed in [33,34,35]), as well as potential treatment strategies (reviewed in [36,37,38,39]). Yet, the aforementioned gold standard for skin cancer diagnostics, such as visual examination and biopsy [40], is still heavily relied on. Jankovskaja et al. [4] explored monitoring of the Trp/Kyn ratio as a potential non-invasive method for detecting skin cancer [4,41]. Although the study solely involved healthy subjects and the simulation of skin cancer conditions using simplified three-dimensional (3D) cell-cultured skin models missing melanocytes or melanoma cells, monitoring the Trp/Kyn ratio proved to be effective and provided valuable insights. The most straightforward way by which small molecules traverse the skin barrier is via passive diffusion. As the first approximation, this process is dictated by the concentration gradient, which, in most cases, drives molecular diffusion from regions of high concentrations to regions with lower concentrations [42]. *Stratum corneum*, (SC), the outermost layer of the skin, constitutes a critical barrier for molecular transport, restricting the diffusion of high-molecular-weight biomarkers from internal compartments towards the skin surface [43]. Nonetheless, both Trp and Kyn are LMW molecules, which can easily cross the SC and reach the skin surface within hours [3]. Given that the overexpression of *IDO-1* and alterations in the Trp/Kyn ratio are observed across all stages of the disease [11,12,13,14], this presents an intriguing direction for employing these potential biomarkers in future melanoma diagnostics. A graphical visualization of the introduction is presented in Figure 1.

In this study, we explored the relevance of monitoring Trp and Kyn levels on the surface of ex vivo reconstructed 3D melanoma (mm) and 3D melanocyte (mc) models, developed in-house. We investigated this sampling method concerning the effect of IFN-γ or UVB on *IDO-1* upregulation in the KP and the resulting alterations in the Trp/Kyn ratios in healthy and disordered 3D skin models. Moreover, to increase the reliability of the analytical method, non-invasive sampling of Trp and Kyn was conducted in parallel with the sampling of two other amino acids, i.e., phenylalanine (Phe) and tyrosine (Tyr). The inclusion of these two amino acids was motivated by the fact that they are found in similar amounts as Trp as a part of the natural moisturizing factor (NMF) pool of the SC [4,44]. Since Kyn is a derivative of Trp and not a native constituent of the NMF, the concentrations of Kyn on the skin surface are much lower than those of Trp, Phe, and Tyr [4,44]. Therefore, monitoring Phe and Tyr alongside Trp served as an additional means of quality control for detecting and reducing potential errors in Trp and Kyn surface sampling and quantification.

## 2. Materials and Methods

### 2.1. Cell Culture and 3D Melanocyte and Melanoma Models Preparation

Dermal Primary Fibroblast cell line neonatal, BJ (ATCC; Manassas, VA, USA; CRL-2522), was grown in EMEM (ATCC) with 1% penicillin/streptomycin (PEST) (Gibco; Waltham, MA, USA) and 10% fetal bovine serum (FBS; heat inactivated) (Thermo Fisher Scientific; Waltham, MA, USA). Primary Human Epidermal Keratinocytes, neonatal, HEKn (Thermo Fisher Scientific; C0015C), were grown in EpiLife with 60 μM calcium (Thermo Fisher Scientific), supplemented with Human Keratinocyte Growth Supplement (HKGS) (Thermo Fisher Scientific) and 0.2% gentamicin sulfate (Corning; Corning, NY, USA). Primary Epidermal Melanocytes, adult, HEMa (ATCC; PCS-200-013), were grown in Dermal Cell Basal Medium (ATCC) supplemented with the Adult Melanocyte Growth Kit (ATCC). The melanoma cell line, Mel Ho (DSMZ; Braunschweig, Germany; ACC 62), was grown in DMEM with 1 mg/mL glucose (Thermo Fisher Scientific), 10% FBS, and 1% PEST. All cells were grown under standard growth conditions (37 °C, 5% CO_2_).

The 3D models were prepared in-house following a modified protocol described by Zoio et al. [45]. The Alvetex™ Scaffold 12 Well Inserts (Reprocell; Beltsville, MD, USA) were used to establish human dermal equivalents containing the dermal fibroblasts (BJ). Before use, the polystyrene scaffolds were activated according to the manufacturer’s protocol, and the inserts were placed into a 12-well plate (VWR International; Radnor, PA, USA). In brief, fibroblasts were detached using Accutase solution (Gibco; Waltham, MA, USA) and counted, and 1.0 × 10^6^ cells were seeded onto each polystyrene scaffold. After 2 h incubation, the inserts were submerged in EMEM with 1% PEST and 10% FBS and supplemented with 100 μg/mL L-ascorbic acid 2-phosphate (Sigma-Aldrich; Burlington, MA, USA). The medium was changed every other day for 14 days. After 14 days, 5.0 × 10^4^ melanoma cells (Mel Ho) or 5.0 × 10^4^ primary melanocyte cells (HEMa) were seeded on top of the dermal equivalents in the inserts followed by 2 h incubation. The inserts were then submerged in DMEM with 1 mg/mL glucose, 10% FBS, and 1% PEST or Dermal Cell Basal Medium supplemented with an appropriate growth kit for 24 h. Next, 5.0 × 10^5^ primary epidermal keratinocytes (HEKn) were seeded on each scaffold and incubated for 2 h. Then, the mc models were submerged in 80% EpiLife supplemented with HKGS and 0.2% gentamicin sulfate, and mixed with 20% Dermal Cell Basal Medium supplemented with the Adult Melanocyte Growth Kit. Meanwhile, the mm models were submerged in 80% EpiLife supplemented with HKGS and 0.2% gentamicin sulfate, combined with 20% DMEM with 1 mg/mL glucose, 10% FBS, and 1% PEST. Both media were freshly supplemented with 50 μg/mL L-ascorbic acid 2-phosphate prior to use, and the 3D models were incubated for 3 days. This was followed by lifting the inserts, realizing an air–liquid interface (ALI) during which the 3D models were grown in 80% EpiLife medium supplemented with HKGS and 0.2% gentamicin sulfate, mixed with 20% DMEM with 1 mg/mL glucose, 10% FBS, 1% PEST (mm model), or 20% Dermal Cell Basal Medium supplemented with the Adult Melanocyte Growth Kit (mc model), and freshly supplemented with 50 μg/mL L-ascorbic acid 2-phosphate, 1.5 mM CaCl_2_ (Alfa Aesar; Ward Hill, MA, USA) and 10 ng/mL recombinant human keratinocyte growth factor (KGF) (R&D Systems; Minneapolis, MN, USA) for 14 days to achieve fully differentiated mc or mm human skin equivalents. A simplified visualization of the mc and mm model preparation is presented in Section 1 (S1) of the Appendix A.

### 2.2. Stimulation of 3D Models

The completed mc and mm models were then divided into three groups. Each group consisted of a negative control (Ctrl), IFN-γ stimulated models (20 ng/mL), and UVB-exposed models at 80 mJ/cm^2^. The UVB exposure was performed using a UV lamp (Analytik Jena; Jena, Germany; model UVP 95-0104-02), with a dose of 80 mJ/cm^2^ (wavelength: 302 nm), corresponding to a full day’s exposure to the sun [46]. The UVB dose was calculated using Equation (1) [47].
(1)$$UVB\;dose\left[\frac{mJ}{{cm}^{2}}\right]=UV\;intensity\left[\frac{\mu W}{{cm}^{2}}\right]\times Time\left[s\right]$$

The IFN-γ stimulation lasted 24 h, after which the basolateral samples were collected from the cell culture media, and then stored at −80 °C pending High-Performance Liquid Chromatography (HPLC) analysis. For topical sampling, 500 μL of PBS was applied to the surface of each 3D model and incubated for 1 h before being collected and stored at −80 °C until HPLC and HPLC-Mass Spectrometry (HPLC-MS) analyses were conducted. A simplified visualization of the 3D model stimulation and sampling method is presented in S1 of the Appendix A.

### 2.3. Histological Analysis and Immunohistochemistry of 3D Melanocyte and Melanoma Models

Half of the 3D models of each type (mc vs. mm models) were fixed in 10% formalin for 24 h at room temperature, then dehydrated and embedded in paraffin. The models were consecutively sectioned at 3 μm. Sections were stained with hematoxylin–eosin (H&E) according to the standard protocol or immunolabeled with antibodies against Melan-A and *IDO-1* as described below.

To detect *IDO-1* and visualize melanocytes or melanoma cells in mc and mm models, the tissue sections (3 μm) were deparaffinized, rehydrated, and then subjected to heat-induced antigen retrieval in citrate buffer (10 mM; pH 6) at 95 °C for 40 min in a Decloaking ChamberTM (NxGe, Biocare Medical; Pacheco, CA, USA). Next, the non-specific background staining was blocked with Background Punisher (MACH4 kit, BRI4012L; Biocare Medical) followed by overnight incubation with primary antibodies diluted with antibody diluent (S2022; DAKO; Glostrup, Denmark). The optimal antibody dilutions were as follows: monoclonal mouse anti-human *IDO-1* (Clone 998743, MAB60301; R&D Systems) 1/2000 and monoclonal mouse anti-human Melan-A (Clone A103, M7196; DAKO) 1/400. Next, endogenous peroxidase was blocked (Peroxidase-Blocking Solution, S2023; DAKO) prior to incubation with a mouse probe (MACH4; Biocare Medical) and secondary antibody (goat anti-rabbit-HRP, MACH4; Biocare Medical). The immunoreaction was visualized with diaminobenzidine (DAB; MACH4; Biocare Medical). The tissue sections were then counterstained with hematoxylin, dehydrated, and mounted. After each step, tissue sections were washed with Tris-buffered saline containing 0.1% Tween 20 (Thermo Fisher Scientific). The tonsils were used as a positive control for *IDO-1* and Melan-A expression, while the N-Universal mouse negative control (N1698; DAKO) served as a negative control.

### 2.4. RNA Extraction, cDNA Synthesis and RT-qPCR

The remaining half of the mc and mm models were dissected from the scaffolds, transferred to individual 1.5 mL Eppendorf tubes, and stored at −80 °C until further analysis. RNA extraction was carried out on the whole 3D models following the manufacturer’s protocol using the RNeasy Mini Kit (Qiagen; Hilden, Germany). Seven hundred microliters of lysis buffer combined with 7 μL of β-mercaptoethanol (Honeywell Fluka; Charlotte, NC, USA) were added to each Eppendorf tube containing the Alvetex membrane with the mc or mm model and pipetted vigorously. Next, each lysate was transferred to the fresh ISOLATE II filter collection tube and centrifuged at 11,000× *g* for 1 min. To mediate RNA binding to the column, 700 μL of 70% ethanol was added to each sample, and the remaining procedure was carried out according to the protocol provided by the manufacturer.

cDNA synthesis was carried out according to the protocol provided by the manufacturer using the SensiFast cDNA synthesis kit (Bioline; London, UK). The mastermix for each sample contained 10 μL of nuclease-free water (BioNordika; Solna, Sweden), 5 μL of isolated RNA, 4 μL of TransAmp Buffer (Bioline), and 1 μL of reverse transcriptase (Bioline). The procedure was run on a VWR thermocycler (Avantor; Radnor Township, PA, USA) with the following program: 25 °C for 10 min, 42 °C for 15 min, and 85 °C for 5 min. The samples were stored at −20 °C.

Quantitative reverse transcription PCR (RT-qPCR) was performed with the use of the SYBR Green Technology kit (Thermo Fisher Scientific), where each sample contained 10 μL of SYBR mix (Sensifast; Bioline), 3 μL of nuclease-free water (BioNordika), 5 μL of template cDNA, 1 μL of forward primer (Invitrogen; Waltham, MA, USA), and 1 μL of reverse primer (Invitrogen). The final concentrations of primers were 0.5 μM. The procedure was then run on the LC480 Lightcycler (Roche; Basel, Switzerland), using the following program: 95 °C for 10 min, 45× cycle at 95 °C during 10 s for melting, 65 °C for 10 s annealing, and 72 °C for 11 s extension. *GAPDH* was used as the reference gene in the procedure [48]. The sequences of the forward and reverse primers used during the RT-qPCR procedure can be found in Table 1.

The relative gene expression based on RT-qPCR was calculated using the ΔΔ-CT-method for each sample [45]. The baseline expression from unstimulated 3D models is represented as “1” in the figures.

### 2.5. Quantification of Tryptophan (Trp), Kynurenine (Kyn), Tyrosine (Tyr), and Phenylalanine (Phe)

Trp and Kyn quantities in basolateral cell culture samples were measured using HPLC connected to a photodiode array (PDA) detector. Trp and Kyn quantification in both basolateral and topical samples and Tyr and Phe quantification in topical samples were performed using HPLC-MS. A description of the preparation procedure of the stock solutions and calibration standards can be found in Section 2 (S2) of the Appendix A.

#### 2.5.1. HPLC-PDA Analysis

To remove the proteins in basolateral cell culture samples, 10 kDa molecular weight cut-off filters (MWCO, PES modified; VWR International; Radnor, PA, USA) were used before Trp and Kyn analysis. A maximum of 500 µL of the sample was pipetted into the filter and centrifuged at 10,000–14,000× *g* at 10 °C for 15 min or until all of the sample solution went through the filter. The quantification of Trp and Kyn in cleaned-up samples was performed using an Agilent HPLC system equipped with a G1315 PDA detector, a G1313A autosampler, a G1316A column oven, a G1322-A in-line degasser, and a G1312A binary pump (Agilent 1100 Series; Agilent Technologies; Waldbronn, Germany). Chromatographic separation was performed on a reversed-phase Kromasil C18 column (250 × 4.6 mm, 5 μm particle size, 100 Å; ES Industries; West Berlin, NJ, USA). Solvents A (10 mM of NaH_2_PO_4_ adjusted to pH 2.8) and B (methanol) were used to create a linear 17 min gradient profile to elute analytes. The flow rate was set to 0.9 mL/min and the column compartment was heated to 40 °C. The separation profile was as follows: 25% of solvent B was held for 7 min, increased to 95% B in 4 min, and held for 4 min; then solvent B was decreased to 25% in 0.1 min and kept at 25% for an additional 1.9 min. Trp and Kyn were monitored at 280 nm and 360 nm, respectively.

The concentrations of Trp and Kyn in the samples were determined using the standard calibration curve approach. The calibration curves were prepared in the range of 0.78 to 100 μM (R^2^ = 0.99). Automatic peak area integration was performed by OpenLAB software (Lab Advisor Basic Software; Agilent Technologies) followed by manual peak inspection. The limit of detection (LOD) was determined to be 0.2–0.4 µM for Kyn and 0.4–0.7 µM for Trp for mc and mm model samples, respectively. The limit of quantification (LOQ) was determined to be 0.5–0.7 µM for Kyn and 1.1–2.1 µM for Trp for mc and mm model samples, respectively. The LODs and LOQs for the mc and mm models differed slightly due to the difference in the type of cell culture medium used for growing these models. The LODs and LOQs were calculated from standard calibration curves as presented in Equations (2) and (3).
(2)$$\mathrm{LOD}=\frac{3.3\sigma }{slope}$$
(3)$$\mathrm{LOQ}=\frac{10\sigma }{slope}$$

σ is the standard error of the y-intercept from the regression analysis of the calibration standards. All the samples, including the standards, were run in triplicate. The precision and accuracy of the HPLC-PDA method can be found in Appendix A, S2).

#### 2.5.2. HPLC-MS Analysis

HPLC-MS analysis of Tyr, Phe, Trp, and Kyn was carried out on an HPLC system (Alliance 2695; Waters; Milford, MA, USA) connected to a Waters micromass ZQ mass spectrometer equipped with an electrospray ion source. The separation of analytes was performed on a reversed-phase Kromasil C18 column (250 × 4.6 mm, 5 µm particle size, 100 Å; ES Industries). Solvents A (0.1% formic acid in water) and B (0.1% formic acid in methanol) were used to create the gradient to elute the analytes. The elution profile at a flow rate of 0.8 mL/min was performed as follows: 10% B was kept for 7 min and then increased to 95% B in 8 min and held at 95% B for 10 min. After that, solvent B was decreased to 10% over 0.1 min and held at 10% for 4.9 min. The quantification of analytes was performed in selected ion reaction (SIR) mode while operating at positive polarity (Appendix A, S2). The capillary voltage was set to 3.5 kV. The cone voltage was set to 20 eV, the extractor at 3 V, and the RF lens at 0.2 V. The source temperature was set to 120 °C. The flow of desolvation gas was 800 L/h, the cone gas flow was set to 25 L/h, and the desolvation temperature was 400 °C. The inter-channel and inter-scan delays were 0.02 and 0.1 s, respectively. Dwell times were 0.25 s for all selected ions. The span window was set to 1 Da.

Data analysis was performed by using MassLynx V4.1 software (Waters). The concentrations of analytes were calculated based on the calibration standards in the range of 0.15 to 5 µM for Phe, Trp, and Kyn, and from 0.45 to 15 µM for Tyr (R^2^ > 0.99). The LODs for topical samples were as follows: 0.3 µM for Tyr, 0.01 µM for Phe, 0.1 µM for Trp, and 0.04 µM for Kyn. The LOQs were 0.9 µM for Tyr, 0.03 µM for Phe, 0.3 µM for Trp, and 0.1 µM for Kyn. The LODs and LOQs were calculated as described in Section 2.5.1. Precision, accuracy, LODs, and LOQs for topical samples measured with HPLC-MS can be found in Appendix A, S2).

### 2.6. Statistical Analysis Using Quantitative Methods

In this study, each biological replicate (*n* = 3) consisted of mc and mm models subjected to three conditions (Ctrl, IFN-γ, and UVB). In each stimulation condition, 3D skin models were prepared in two technical replicates (two 3D models per condition; *n* = 3 × 3 × 2 = 18 inserts of mc models and a corresponding number of mm models were generated in this study; hence, 36 inserts of 3D skin models in total). All measurement data are expressed as mean ± standard deviation (SD), derived from the indicated number of independent experiments. All experiments were repeated at least three times (unless stated otherwise) as indicated in each figure. The significance of any difference was determined using two-way ANOVA followed by Dunnett’s multiple comparisons or Tukey multiple comparisons. Statistical analyses were performed using GraphPad Prism, version 10.1.0 (GraphPad Inc.; San Diego, CA, USA). Scatter plots illustrating the relationship between Kyn and Trp concentrations across different treatments (Ctrl, IFN-γ, and UVB) and batches (1–3) with added t-distribution ellipses were made by using R version 4.3.1 (R Foundation for Statistical Computing; Vienna, Austria) ggplot2 package. A *p*-value of < 0.05 (*p* < 0.05) was considered statistically significant. * *p* ≤ 0.05 ** *p* ≤ 0.01 *** *p* ≤ 0.001.

### 2.7. Ethical Considerations

The cellular components used to create the 3D models were obtained from well-established cell repositories, specifically the American Type Culture Collection (ATCC) and the Leibniz Institute DSMZ-German Collection of Microorganisms and Cell Cultures (DSMZ). Therefore, given that both the primary cells and cell lines were sourced from reputable suppliers and the study did not involve human subjects, no ethical approval was required in this research project.

## 3. Results

### 3.1. IFN-γ Upregulates IDO-1 Expression in Melanoma and Melanocyte Models, While UVB Exposure Upregulates IL-6 and IDO-1 Expression in Both Models

RT-qPCR was used to monitor how IFN-γ and UVB affect *IL-6* and *IDO-1* expression in the 3D models.

Exposure to 20 ng/mL IFN-γ for 24 h had no significant effect on *IL-6* expression in either of the models (Figure 2a). In contrast, UVB exposure led to a significant upregulation of *IL-6* in the mm model (* *p* < 0.05). Specifically, in the mc model, *IL-6* exhibited a twofold increase, while in the mm model, the level of *IL-6* was five times higher than that observed in the control group. Exposure to IFN-γ resulted in a notable increase in the amount of *IDO-1*, with the increase in expression levels remarkably similar for both models relative to control tissues (Figure 2b). In contrast, exposure to UVB resulted in a moderate increase in *IDO-1* levels, with the increase being more pronounced in the mm model.

### 3.2. Upon IFN-γ Stimulation, IDO-1 Protein Expression Is Detectable in the Fibroblasts of the Melanocyte Model, and in the Fibroblasts as Well as Some Melanoma Cells of the Melanoma Model

After discovering that IFN-γ and UVB increased *IDO-1* levels in both models, albeit to varying extents, the objective was to investigate whether this disparity caused any differences in tissue morphology and localization of *IDO-1* protein expression in the respective models.

Therefore, to ensure that the mc models closely mimicked the anatomical structure of native human skin, H&E staining was employed for visualization (Figure 3a,d,g). In all testing conditions, the cross-sectional analysis of the mc model revealed a well-structured and fully differentiated epidermal layer. Moreover, a fibroblast-enriched layer was observed within the dermal compartment, albeit with occasional keratinocyte infiltration into the scaffold material (Figure 3g). Subsequently, targeting and visualizing the melanocyte cells within the basal layer involved the utilization of immunohistochemical staining (IHC) for Melan-A (Figure 3b,e,h). The *IDO-1* staining unraveled distinctive patterns of *IDO-1* protein expression within tissues subjected to the different treatment conditions. In IFN-γ stimulated models, *IDO-1* expression was evident in fibroblasts, while there was no detectable *IDO-1* observed in melanocytes (Figure 3f). Conversely, analogous to the control tissue (Figure 3c), in UVB-stimulated models, no *IDO-1* was detected in any cells of the tissue (Figure 3i).

Likewise, in parallel with the observations in the mc model, the cross-sections of the mm model, stained with H&E, exhibited good stratification and differentiation within the epidermis (Figure 4a,d,g). The dermal regions displayed an even distribution of fibroblasts; however, there was a subtle difference in confluence compared to the mc model, i.e., the cell density was slightly higher in the mm models. Moreover, the cross-sectional Melan-A staining under various testing conditions yielded interesting results. In the control (Ctrl) (Figure 4b) and upon IFN-γ stimulation (Figure 4e), melanoma cells were highly localized and formed melanoma nests within the epidermis of the mm model. Since the melanoma cell line in the mm model originates from a primary melanoma (i.e., melanoma in situ), this accounts for the absence of visible melanoma cell penetration into the scaffold material. Simultaneously, in UVB-stimulated models, the melanoma nests were absent from the tissue (Figure 4h). Next, IHC staining for *IDO-1* was employed to identify the specific cell types displaying increased expression of *IDO-1*. The staining revealed that following IFN-γ stimulation, noticeable localization of *IDO-1* was present in both fibroblasts and some melanoma cells in the melanoma nests (Figure 4f), relative to the control (Figure 4c). In contrast, no detectable traces of *IDO-1* were found following UVB stimulation (Figure 4i), aligning with the findings from RT-qPCR.

### 3.3. IFN-γ Stimulation Causes Depletion in Trp and an Increase in Kyn Concentrations in Basolateral and Topical Samplings

Given that the absolute Trp concentrations in the cell culture media are fixed, the assumption was made that the quantity of this amino acid would be higher in the basolateral samples from both 3D models, compared to the concentrations obtained from topical sampling of the PBS. Therefore, considering this, Trp and Kyn levels were first measured in the basolateral cell culture media from both models with HPLC-PDA.

Upon stimulation with 20 ng/mL IFN-γ, the mc model exhibited a significant increase in Kyn concentration, accompanied by a significant reduction in Trp concentration (** *p* < 0.01) (Figure 5a). A similar pattern was observed in the mm model, although with a more significant reduction in the Trp level along with a more significant increase in the Kyn concentration (*** *p* < 0.001), (Figure 5b). In contrast, the Kyn concentration increased significantly (* *p* < 0.05) relative to the control in response to UVB stimulation in the mm model (Figure 5b) while being unaffected in the mc model (Figure 5a).

Since the quantification of Trp and Kyn topical concentrations in the PBS samples was the primary focus of this study, the HPLC-PDA method was also employed in these samples. However, given that the Trp and Kyn quantities were sometimes below the LOQ in topical samples from both models, the decision was made to utilize a more sensitive detection method, namely, HPLC-MS.

The general concentration pattern observed for the topical sampling was similar overall to the measured concentrations from basolateral sampling. Specifically, in parallel with the basolateral concentrations, the topical samples also demonstrated Trp depletion and Kyn elevation in response to IFN-γ stimulation. The observed pattern was consistent between both models, albeit with a more significant (*** *p* < 0.001) Trp reduction in the mc model (Figure 6a) and higher absolute topical concentrations detected in the mm model compared to the mc model (Figure 6).

### 3.4. IFN-γ Stimulation Leads to a Modest Increase in Tyr and Phe in the 3D Melanoma Model, While Simultaneously Decreasing Tyr and Phe in the 3D Melanocyte Model

To identify potential additional LMW biomarkers from the 3D skin models and to address possible errors related to the observed Trp and Kyn amounts, the fluctuations in concentration levels between Tyr and Phe were investigated in the topical samples.

Following IFN-γ stimulation, the Phe concentration decreased significantly (* *p* < 0.05) in the mc model compared to the control (Figure 7a). In the mm model, no significant changes could be detected. Concurrently, exposure to UVB radiation did not lead to significant changes in Tyr and Phe concentrations in either model (Figure 7a,b).

### 3.5. The Trp/Kyn Ratio Decreases in Both 3D Models after IFN-γ Stimulation While Phe/Trp and Tyr/Trp Ratios Increase

In addition to evaluating the Trp/Kyn ratio, the decision was made to also investigate the Tyr/Trp, Phe/Trp, and Phe/Tyr ratios, which served as quality controls.

IFN-γ stimulation led to a substantial reduction in Trp/Kyn ratios in both 3D models (Figure 8). While Phe/Trp and Tyr/Trp ratios significantly increased (*** *p* < 0.001) relative to the control under these conditions, the Phe/Tyr ratio remained unchanged. At the same time, UVB stimulation resulted in no alterations in Phe/Trp and Tyr/Trp ratios in either model when compared to the control. In the mc model, the Trp/Kyn ratio showed a modest increase (Figure 8a), while in the mm model, the ratio exhibited a notable decrease following UVB treatment (Figure 8b). A quantitative summary of the trends visualized in Figure 8 is provided below in Table 2 and Table 3.

Based on the quantitative data, stimulation with IFN-γ resulted in a remarkable 35-fold reduction in the Trp/Kyn ratio in the mc model, compared to the control (Table 2). Similarly, in the mm model, the same ratio showed a 30-fold decrease following the same IFN-γ stimulation (Table 3). Additionally, the Phe/Trp and Tyr/Trp ratios in the mc model were consistently around 11–12 times higher upon IFN-γ stimulation (Table 2). In contrast, in the mm model, these ratios increased 15–16 times relative to the control (Table 3). In contrast to IFN-γ stimulation, when examining the Trp/Kyn ratios between the two UVB-stimulated models, the differences in ratios became less evident. The Trp/Kyn ratio was found to be 5 times lower in the mm model (Table 3), as opposed to the mc model where the Trp/Kyn ratio remained unchanged relative to the control tissue (Table 2).

## 4. Discussion

In recent decades, there has been a worrying rise in malignant melanoma cases, prompting numerous efforts to understand and address the associated risk factors. Despite accounting for only 5% of skin cancer cases, melanoma is currently the deadliest form of skin cancer [49]. Therefore, the importance of early-stage diagnosis cannot be overstated since it significantly improves the 5-year patient survival rate (94%) [50]. However, the current gold standard diagnostic methods, such as visual inspection and biopsy, have inherent limitations that restrict the efficient detection of melanoma at the early stages of development [51,52]. Thus, establishing a novel, complementary diagnostic approach should benefit both patients and healthcare providers.

In this study, we investigated an alternative non-invasive diagnostic method based on monitoring Trp and Kyn levels on the surface of in-house reconstructed 3D skin models. The models simulated healthy (mc model) and disordered skin with cancer development (mm model). Furthermore, we utilized UVB or IFN-γ to induce the KP as part of this study. The current investigation not only establishes the correlation between *IDO-1* upregulation and the topical Trp/Kyn ratio but also underscores the pivotal role of IFN-γ in the transformation and progression of melanoma.

Immune activation through inflammation and immune inhibition through immunosuppression represent two contrasting facets of the immune response spectrum [53]. However, it is well-established that despite their divergent effects, inflammation and immunosuppression can coexist in the process of cancer development [54,55]. For instance, in melanoma initiation, UVB exposure is widely recognized as an inducer of skin inflammation [56,57,58]. Moreover, prolonged, repetitive UVB exposure can generate a state of chronic inflammation, a critical factor in maintaining the pro-tumoral environment. Typically, the pro-tumoral environment is characterized by the presence of proinflammatory mediators and the accumulation/activation of immune suppressor cells [54]. Since the presence of *IL-6* in the cell environment can induce a positive feedback loop that leads to *IDO-1* upregulation and increased Trp-to-Kyn conversion [26], it can also drive the immunosuppressive processes that occur from depriving the immune cells of Trp (Figure 1). While malignant melanoma is recognized for its ability to hijack and subsequently evade the immune system [59], it also exploits various cellular elements and cytokines generated and released within the TME [60]. Another critical cytokine in this context is IFN-γ, primarily secreted by macrophages and activated T cells [20]. However, when introduced into the TME, its function is reversed, and instead of manifesting its conventional anti-tumor properties, IFN-γ acts as an immunosuppressant [19]. For instance, Zaidi et al. [61] discovered a significant mechanism in melanoma development, in which UVB exposure initiates melanocyte activation by inducing ligands that interact with the C-C chemokine receptor type 2 (CCR2). This interaction, in turn, triggers the activation and recruitment of CCR2+ macrophages, which infiltrate the UVB-exposed neonatal skin. Subsequently, the macrophages begin to secrete IFN-γ into the cell microenvironment leading to melanocyte proliferation, migration, and the expression of various genes involved in immune evasion or cell survival. All these processes are melanomagenic, ultimately facilitating melanoma progression [61].

As previously introduced (Figure 1), during melanoma development, IFN-γ and *IL-6* drive *IDO-1* overexpression through a negative and positive feedback mechanism, respectively [20,26]. Therefore, in this study, we examined the role of these cytokines on *IDO-1* overexpression and the resulting alterations in Trp and Kyn levels, particularly on the surface of the reconstructed 3D skin models. In light of the findings obtained from RT-qPCR, it appears that IFN-γ does not have any observable influence on the *IL-6* expression in the reconstructed 3D models (Figure 2a), while at the same time, it significantly increases *IDO-1* expression levels in both models (Figure 2b). This observation confirms that the negative feedback signal triggered by IFN-γ serves as a driver of *IDO-1* overexpression. Simultaneously, the absence of an increase in *IL-6* expression following IFN-γ stimulation might be attributed to delayed kinetics. In other words, the stimulation of *IL-6* expression driven by Kyn-mediated AhR activation may not have fully manifested within the specified time frame of 24 h. In contrast, UVB exposure leads to a significant increase in *IL-6* expression, particularly in the mm model (Figure 2a). The elevation of *IL-6* expression induced by UVB exposure in these models may indicate the initiation of inflammation not only within malignant cells of the TME but also in healthy cells, as seen in the mc model. At the same time, UVB has a more modest impact on *IDO-1* expression (Figure 2b).

In this study, the highest IFN-γ concentration (20 ng/mL) was in the cell culture medium at the bottom of the inserts. Therefore, we also explored any signs of melanoma progression in the 3D models by performing immunohistochemical staining for *IDO-1* and Melan-A. The latter is a product of the Melanoma Antigen Recognized by T cells 1 (MART-1) gene, a protein expressed by melanocytes and melanoma cells. The observations noted in the cross-sectional analysis of both 3D models suggest that IFN-γ permeates the scaffold material, thus influencing *IDO-1* expression in the fibroblasts (Figure 3f and Figure 4f). Additionally, IFN-γ appears to also have the potential to penetrate the basal membrane of the epidermis, extending its impact to regions occupied by the melanoma cells in the mm model (Figure 4f), thus driving melanoma progression (Figure 4e). Interestingly, after UVB exposure, melanoma nests were not visually detectable in the mm model (Figure 4h). UVB radiation is known to induce apoptosis in cells through various mechanisms [62]. Thus, the lack of visible melanoma nests could likely be attributed to UVB-induced apoptosis within the melanoma cells of these nests.

The analysis of Trp and Kyn concentrations in basolateral and topical samples exposed to IFN-γ revealed a substantial decrease in Trp levels and a concurrent rise in Kyn concentrations (Figure 5 and Figure 6). These results not only align with the findings reported in the existing literature [20] but also correspond with the heightened expression of *IDO-1* in the reconstructed 3D skin models, confirming that IFN-γ drives the negative feedback signal, which leads to *IDO-1* upregulation and increased Trp-to-Kyn conversion (Figure 2b). Similarly, the outcomes of UVB exposure appear to match the results of *IDO-1* expression under identical treatment conditions. Notably, UVB exposure seems to have a minimal impact on Trp-to-Kyn conversion in the mc model (Figure 6a), which pairs with the relatively modest increase in *IDO-1* expression observed in RT-qPCR (Figure 2b). Conversely, in the mm model, UVB exposure appears to cause a more notable increase in Kyn concentrations, while having minimal to no impact on Trp concentrations (Figure 5b and Figure 6b). This more pronounced elevation in Kyn levels could potentially be attributed to the increased permeability of the SC, possibly resulting from the weakened barrier function [63] caused by the formation of the melanoma nests in the mm model (Figure 4e). Furthermore, the absence of *IDO-1* expression in epidermal cells like mature keratinocytes and melanocytes [64,65] can also have an impact on the distribution of small molecules within the models.

Prior research suggested that a reduced Trp/Kyn ratio is linked to lower survival rates in melanoma patients, particularly among cases where *IDO-1* upregulation was present [14,15]. Moreover, Gustafsson et al. [65] reported increased Trp-to-Kyn conversion upon IFN-γ stimulation in a reconstructed human epidermis (RHE). However, the conclusions drawn based on the RHE are not entirely comparable given the absence of other essential epidermal components such as melanocytes, and a fully functional fibroblast-enriched dermal layer [65]. Therefore, in this study, we investigated differences in Trp/Kyn ratios between healthy (mc model) and disordered skin models (mm model) studied ex vivo under different treatment conditions. The 3D models comprised a fully differentiated epidermis, enriched with either melanocytes or melanoma cells, and a fibroblast-enriched dermis. Stimulation with IFN-γ (20 ng/mL) was undertaken to mimic inflammation and the TME, while UVB exposure (80 mJ/cm^2^) was used to simulate an inflammatory response. In the mm models, both IFN-γ stimulation and UVB exposure led to a substantial decrease in the Trp/Kyn ratio compared to control tissues, with 30- and 5-fold reductions in the Trp/Kyn ratios, respectively (Table 3). This observation suggests that inflammatory conditions and a pro-tumor microenvironment drive the KP. However, in the mc model, this effect was observed only after IFN-γ stimulation, with a 35-fold reduction in the Trp/Kyn ratio (Table 2). The results from the mc model are particularly compelling, as they indicate the possibility of identifying increased Trp-to-Kyn conversion before the cells undergo malignant transformation. This underscores the clinical significance of using skin surface Trp/Kyn ratio measurements to monitor skin inflammation and cancer.

However, a previous study by Morin et al. [41] indicated that relying solely on skin surface monitoring of the Trp/Kyn ratio as the only cancer biomarker may be inadequate. Therefore, in this study, we decided that the inclusion of Phe and Tyr in the sampling process could offer a more robust approach, drawing on the fact that both amino acids are components of the NMF and thus are present in amounts similar to Trp in the SC. By incorporating a broader spectrum of amino acids in this study and identifying a set of biomarkers that could serve as quality control, we aimed to reduce the sensitivity of the Trp/Kyn ratio and minimize potential errors in the surface sampling and quantification of Trp and Kyn. Similarly to the study conducted by Jankovskaja et al. [4], we observed that a reduction in Trp concentrations due to IFN-γ stimulation resulted in significant alterations in Tyr/Trp and Phe/Trp ratios in both 3D skin models (Figure 8a,b). However, unlike the Trp/Kyn ratio, which decreased under these conditions, the Tyr/Trp and Phe/Trp ratios increased upon IFN-γ stimulation. In the mc and mm models, the Phe/Trp ratios increased 11–16 times upon IFN-γ stimulation, while the Tyr/Trp ratio increased 12–15 times in both models (Table 2 and Table 3). On the other hand, UVB stimulation did not affect Phe/Trp and Tyr/Trp ratios in either model. Consistent with the findings from the study on healthy volunteers [4], we report that analyzing concentrations of the individual amino acids can be used to identify ratios that are sensitive to inflammation and cancer, such as the Trp/Kyn ratio. However, determining ratios that are unaffected by inflammation and cancer, such as the Phe/Tyr ratio, can be instrumental for quality control purposes.

One clear limitation of the current study is the challenge in precise quantification of the changes in Trp, Kyn, Tyr, and Phe levels due to inherent variances among the 3D skin models. These variations may be attributed to batch effects arising from differences between various cell batches [66], and the differences between cell batches can arise from various factors, including variations in the source of the cells, passage number, culture conditions, and other experimental variables [67]. The inherent cell viability can also introduce alterations in cell metabolic activities [68,69], which in turn can influence factors such as the structure of the epidermis and the level of gene expression at any given point. While reconstructing the 3D skin models, along with commercially available cell lines (Mel Ho and BJ), we also utilized the primary cells (HEKn and HEMa). Given that the batch variations are an inherent part of working with commercially available primary cells originating from different donors, as well as the fact that cell lines expanded over a prolonged period, it is important to acknowledge such risks. Consequently, while these variations may introduce some level of uncertainty regarding the results, they reflect the practical challenges faced in experimental research and the need for robust statistical analysis to account for these differences. In this study, we have taken measures to ensure data consistency and rigor in the analyses, aiming to derive meaningful insights despite the inherent variances encountered. In a manner parallel to the variabilities observed among healthy volunteers [4], the 3D models enable us to predict the overall trend of concentration or ratio changes (whether they increase or decrease) rather than provide precise quantitative values with strong statistical significance. This is illustrated in Appendix A as scatter plots outlining the relationship between the Kyn and Trp concentrations in topical samplings from the mc model (Appendix A) and basolateral samplings from the mc (Appendix A) and mm models (Appendix A). The plotted data reveal inherent variations among samples from distinct technical and biological replicates. However, it is worth noting that the overall trend remains consistent across different treatment conditions. Therefore, such observations underscore the importance of interpreting the results in terms of broader trends and patterns rather than specific numerical values.

Nevertheless, the increased conversion of Trp into Kyn, effectively lowering the Trp/Kyn ratio, has been shown to lead to immunosuppressive effects [9,20]. However, it is important to note that the 3D skin models generated for this study lack the immune components. Therefore, it is essential to recognize that certain facets of the KP, specifically the impact of IFN-γ, UVB, and *IL-6* on immune-related aspects, remain unexplored within the scope of this work (Figure 1).

Despite these limitations, the 3D skin models presented in this study seem to be appropriate since they provide a more controlled environment for Trp-to-Kyn topical sampling in comparison to in vivo skin [4,41]. This is particularly attributed to the absence of sweat glands in 3D skin models, which eliminates the interference on sampling caused by sweating. Moreover, the absence of skin microbiota further contributes to the stability of Trp and Kyn concentrations on the surface of the reconstructed 3D models [3]. The models also provide an opportunity to conduct experiments that would be considered ethically questionable in human research. For instance, subjecting the models to controlled doses of UVB radiation or IFN-γ over specific periods helps us gain a deeper insight into the involvement of the KP in the progression of melanoma. This invaluable knowledge can be obtained without ethical concerns or risk to human subjects when studied in 3D skin models.

In future studies, investigating the synergistic or antagonistic effects of the combined IFN-γ and UVB exposure on 3D skin models could be a valuable research avenue. This approach might provide insights into the complex interplay between environmental exposures and immune responses in melanoma development and progression. Such a combination would also more accurately reflect real-world scenarios where patients with early-stage melanoma have recently been exposed to UVB.

Moreover, it is crucial to consider the inclusion of immune components within the reconstructed 3D skin models. For example, the addition of Langerhans cells incorporated in the epidermal compartment of the models would enable a comprehensive exploration of potential immune suppressive effects resulting from elevated Trp-to-Kyn conversion (Figure 1), providing a more accurate understanding of the intricate effects that emerge from *IDO-1* overexpression during melanoma initiation and progression.

Additionally, it is well known that primary melanomas can arise from different driving mutations, such as those in *BRAF* or *NRAS* [70]. These genetic differences contribute to the biological variability of primary melanomas and influence their metabolic footprint. Therefore, future studies could benefit from replacing the currently used melanoma cell line, Mel-Ho, which exhibits a *BRAF* mutation, with one possessing an *NRAS* mutation, such as Mel Juso. However, our previous research comparing primary adult melanocytes and six melanoma cell lines—derived from both primary and metastatic sources—in monolayers did not show significant differences in *IDO-1* expression levels [71].

Furthermore, considering that the melanoma cell line integrated into the mm model was derived from a primary melanoma (melanoma in situ), it would be intriguing to conduct a comparative analysis with an mm model that replicates more advanced melanoma characteristics (malignant melanoma), such as the potential to metastasize [72]. This comparison could provide valuable insights into the variations between these two distinct stages of melanoma development. Understanding the variations in Trp and Kyn concentrations between these stages as well as their impact on the collective events that emerge from the KP is essential to shedding light on the mechanisms contributing to melanoma progression. Moreover, it may help in the identification of novel potential biomarkers for distinguishing between in situ and malignant melanomas, further enhancing the accuracy of melanoma diagnosis and prognosis.

In addition, to the best of our knowledge, this study is the first to utilize topical non-invasive sampling with ex vivo 3D skin models. This method not only provides a more authentic representation of non-invasive sampling compared to traditional cell culture medium sampling but also opens new possibilities for refining non-invasive sampling techniques. For example, future research could explore topical sampling using alternative methods such as hydrogels, cubic liquid crystals, or tape, potentially allowing an analysis of a broader range of analytes [41]. Additionally, there could be opportunities to optimize this approach by determining the most efficient sampling times.

Looking ahead, enhancing reconstructed 3D skin models, e.g., by incorporating immune cells, could create even more reliable tools for exploring new aspects of the KP in vitro. This enhancement could expand our understanding of the pathway’s role in melanoma initiation and progression. Additionally, the insights gained from monitoring Trp and Kyn levels underscore the need for further studies regarding the clinical relevance of non-invasive topical sampling of skin cancer biomarkers. Furthermore, this approach could also lead to research on other potential skin cancer biomarkers, utilizing changes in their ratios, in conjunction with Trp, as predictors of inflammation and cancer.

## 5. Conclusions

To conclude, in this study, Kyn was produced by Trp catabolism mediated by *IDO-1* overexpression in the IFN-γ stimulated 3D models of healthy and disordered skin. Exposure of both 3D skin models to IFN-γ resulted in a reduction in Trp and an increase in Kyn concentrations. The topical ratio of Trp/Kyn was reduced 30–35 times in both 3D skin models upon IFN-γ-stimulation, while upon UVB exposure, only the mm model showed a reduction in the Trp/Kyn ratio. Therefore, 3D skin models can serve as functional tools for studying parts of the KP and its role in early melanoma detection and cancer development. Overall, the findings of this study offer significant promise for advancing the current landscape of early melanoma diagnostics.

## Figures and Tables

**Figure 1 biomolecules-14-00815-f001:**
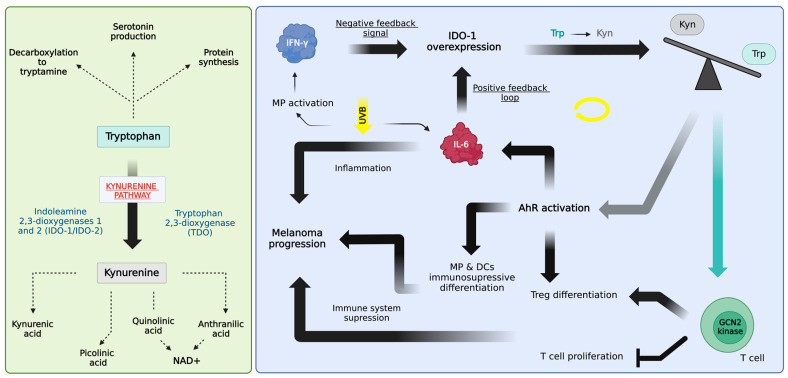
Graphical visualization of the introduction. (**Left side**) A simplified diagram of tryptophan’s role in major metabolic pathways with a primary focus on the Kynurenine Pathway. (**Right side**) The roles of IFN-γ, *IL-6*, and UVB in the upregulation of the Kynurenine Pathway and melanoma progression. IFN-γ—interferon gamma; *IDO-1*—indoleamine 2,3-dioxygenase 1; Trp—tryptophan; Kyn—kynurenine; GCN2 kinase—general control nonderepressible 2 kinase; AhR—aryl hydrocarbon receptor; *IL-6*—interleukin 6; MP—macrophages; DCs—dendritic cells. Created with BioRender.com (accessed on 6 May 2024).

**Figure 2 biomolecules-14-00815-f002:**
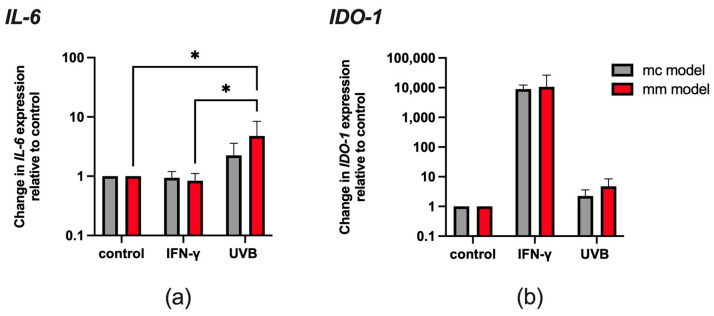
Effect of IFN-γ stimulation or UVB radiation exposure on the expression of *IL-6* (**a**) and *IDO-1* (**b**) in mc and mm models. Both models were stimulated with 20 ng/mL IFN-γ or exposed to 80 mJ/cm^2^ UVB radiation before RNA extraction was performed. Data are shown as the mean ± SD of three independent experiments, * *p* < 0.05 compared to untreated cells (two-way ANOVA followed by Tukey multiple comparisons).

**Figure 3 biomolecules-14-00815-f003:**
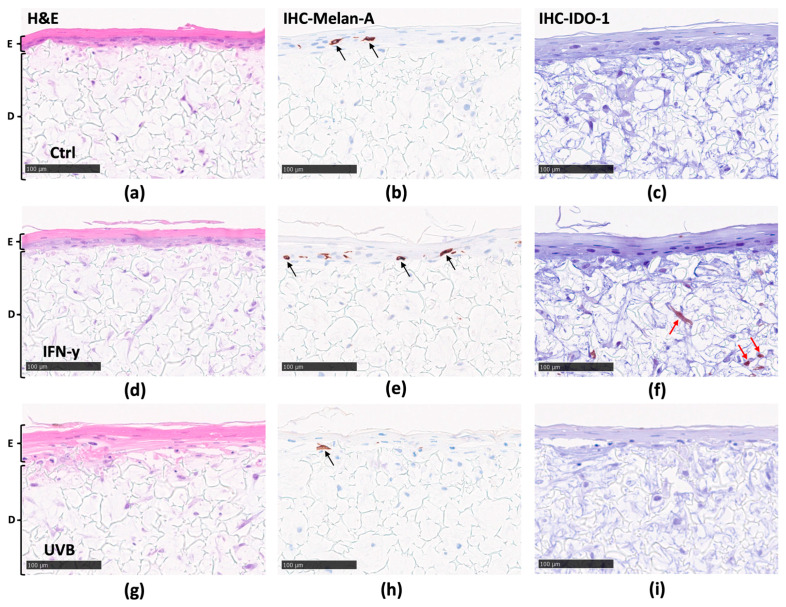
Mc model. Representative sections of mc model exposed to IFN-γ or UVB and stained with H&E (**a**,**d**,**g**). Identification of melanocytes with melanoma antigen recognized by T cells (Melan-A) antibody (**b**,**e**,**h**). Expression of *IDO-1* (**c**,**f**,**i**). One representative of three (**a**–**i**) experiments is shown. Scale bars are 100 μm. E—epidermis; D—dermis; Ctrl—control. Black arrows indicate melanocytes stained for Melan-A. Red arrows indicate fibroblasts where *IDO-1* is localized.

**Figure 4 biomolecules-14-00815-f004:**
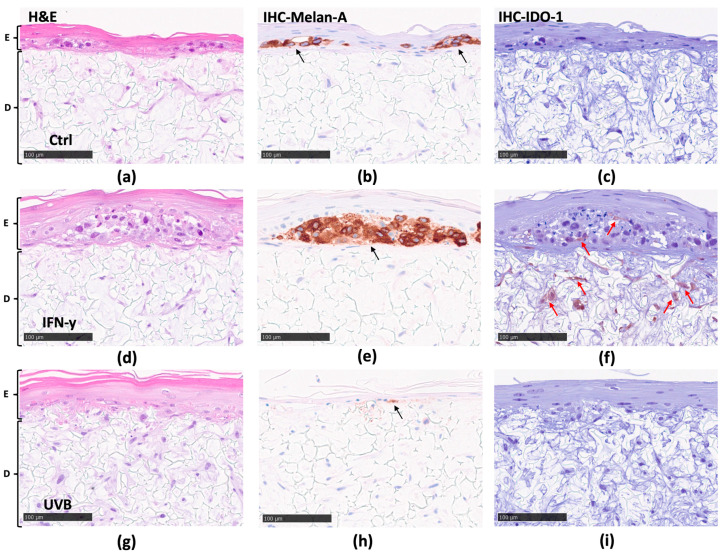
Mm model. Representative sections of mm model exposed to IFN-γ or UVB and stained with H&E (**a**,**d**,**g**). Identification of melanocytes with melanoma antigen recognized by T cell (Melan-A) antibody (**b**,**e**,**h**). Expression of *IDO-1* (**c**,**f**,**i**). One representative of three (**a**–**i**) experiments is shown. Scale bars are 100 μm. E—epidermis; D—dermis; Ctrl—control. Black arrows indicate melanoma cells stained for Melan-A. Red arrows indicate fibroblasts and some melanoma cells where *IDO-1* is localized.

**Figure 5 biomolecules-14-00815-f005:**
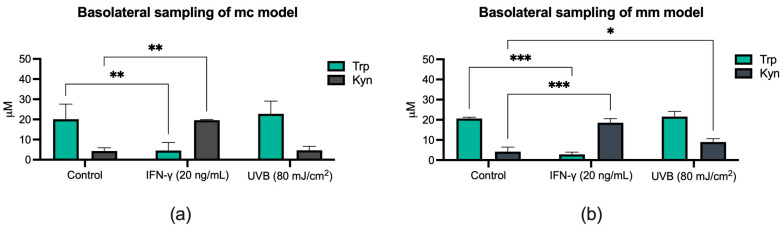
Effect of IFN-γ or UVB on extracellular concentrations of tryptophan (Trp) and kynurenine (Kyn) in basolateral samples from mc (**a**) and mm (**b**) models. Both models were stimulated with 20 ng/mL IFN-γ or exposed to 80 mJ/cm^2^ UVB radiation before samples were collected and analyzed with HPLC-PDA. Data are shown as the mean ± SD of three independent experiments, * *p* < 0.05, ** *p* < 0.01, or *** *p* < 0.001 compared to untreated cells (two-way ANOVA followed by Dunnett’s multiple comparisons).

**Figure 6 biomolecules-14-00815-f006:**
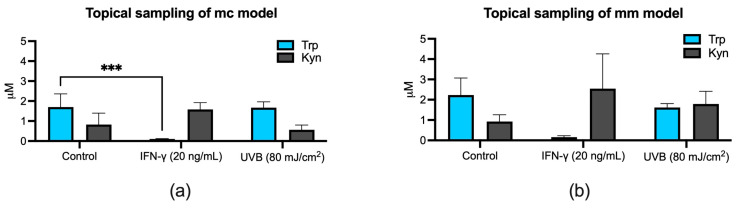
Effect of IFN-γ or UVB on extracellular concentrations of tryptophan (Trp) and kynurenine (Kyn) in topical samples from mc (**a**) and mm (**b**) models. Both models were stimulated with 20 ng/mL IFN-γ or exposed to 80 mJ/cm^2^ UVB radiation before samples were collected and analyzed with HPLC-MS. Data are shown as the mean ± SD of three (**a**) and two (**b**) independent experiments, *** *p* < 0.001 compared to untreated cells (two-way ANOVA followed by Dunnett’s multiple comparisons).

**Figure 7 biomolecules-14-00815-f007:**
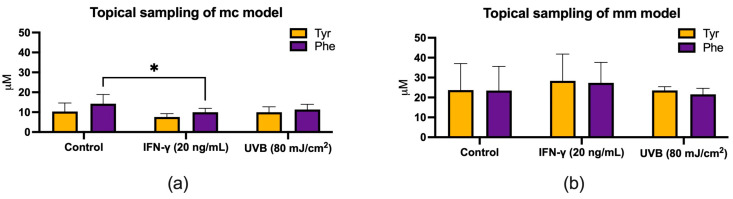
Effect of IFN-γ or UVB on extracellular concentrations of tyrosine (Tyr) and phenylalanine (Phe) in topical samples from mc (**a**) and mm (**b**) models. Both models were stimulated with 20 ng/mL IFN-γ or exposed to 80 mJ/cm^2^ UVB radiation before samples were collected and analyzed with HPLC-MS. Data are shown as the mean ± SD for three (**a**) and two (**b**) independent experiments, * *p* < 0.05 compared to untreated cells (two-way ANOVA followed by Dunnett’s multiple comparisons).

**Figure 8 biomolecules-14-00815-f008:**
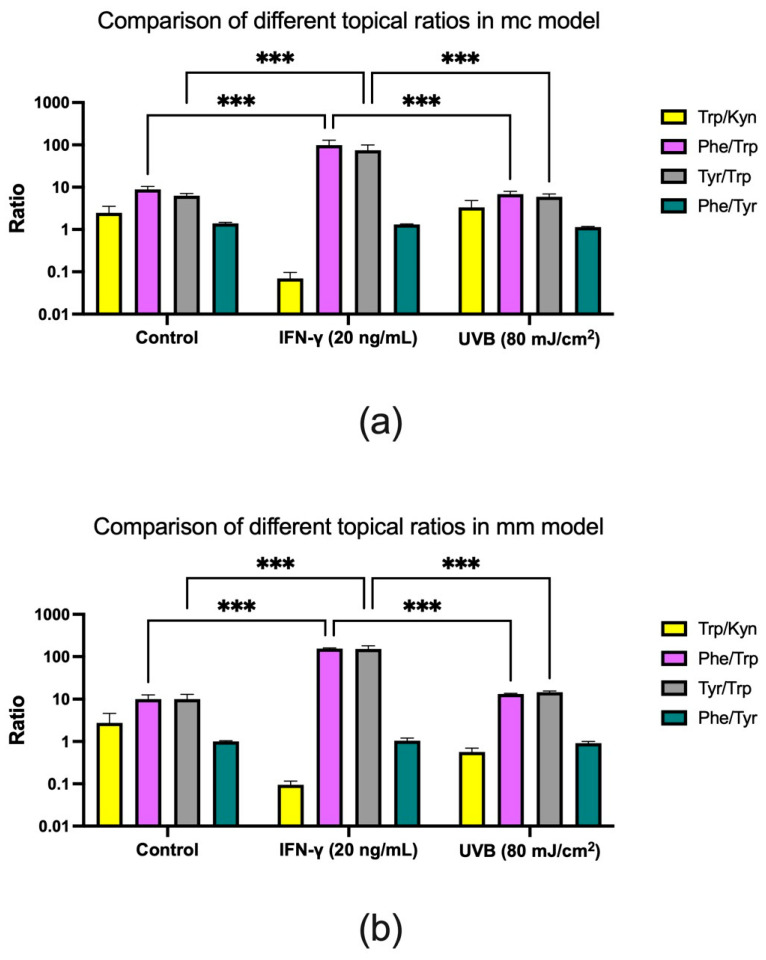
Effect of IFN-γ or UVB on Trp/Kyn, Phe/Trp, Tyr/Trp, and Phe/Tyr ratios in topical samples from mc (**a**) and mm (**b**) models. Both models were stimulated with 20 ng/mL IFN-γ or exposed to 80 mJ/cm^2^ UVB radiation. Data are shown as the mean ± SD for three (**a**) and two (**b**) independent experiments, *** *p* < 0.001 compared to untreated cells (two-way ANOVA followed by Tukey multiple comparisons).

**Table 1 biomolecules-14-00815-t001:** Primers used during quantitative reverse transcription PCR for cDNA amplification.

Gene	Forward Primer	Reverse Primer
*IDO-1*	GAAAGGCAACCCCCAGCTAT	GGAGGAACTGAGCAGCATGT
*IL-6*	AGACAGCCACTCACCTCTTCAG	TTCTGCCAGTGCCTCTTTGCTG
*GAPDH*	AACAGCGACACCCACTCCTC	GGAGGGGAGATTCAGTGTGGT

**Table 2 biomolecules-14-00815-t002:** The effect of IFN-γ or UVB treatment on the mc model. Trp/Kyn, Phe/Trp, Tyr/Trp, and Phe/Tyr ratios were estimated from concentrations of the appropriate amino acids in the topical samples. The ratios were quantified 24 h after starting the IFN-γ treatment. UVB indicates UVB irradiation of the mc model. In this case, the ratios were determined 24 h after the UVB treatment. The arrows indicate changes in the ratio compared to the control, where ↓ signifies a decrease, while ↑ signifies an increase.

Treatment	Mc Model				Ratio Change (×Times) [Compared to Ctrl]			
	Trp/Kyn	Phe/Trp	Tyr/Trp	Phe/Tyr	Trp/Kyn	Phe/Trp	Tyr/Trp	Phe/Tyr
Control	2.5	8.8	6.3	1.4	-	-	-	-
IFN-γ (20 ng/mL)	0.1	98.0	74.9	1.3	≈35↓	≈11↑	≈12↑	≈1
UVB (80 mJ/cm^2^)	3.3	6.8	5.9	1.1	≈1	≈1	≈1	≈1

**Table 3 biomolecules-14-00815-t003:** The effect of IFN-γ or UVB treatment on the mm model. Trp/Kyn, Phe/Trp, Tyr/Trp, and Phe/Tyr ratios were estimated from concentrations of the appropriate amino acids in the topical samples. The ratios were quantified 24 h after starting the IFN-γ treatment. UVB indicates UVB irradiation of the mm model. In this case, the ratios were determined 24 h after the UVB treatment. The arrows indicate changes in the ratio compared to the control, where ↓ signifies a decrease, while ↑ signifies an increase.

Treatment	Mm Model				Ratio Change (×Times) [Compared to Ctrl]			
	Trp/Kyn	Phe/Trp	Tyr/Trp	Phe/Tyr	Trp/Kyn	Phe/Trp	Tyr/Trp	Phe/Tyr
Control	2.7	10.0	10.1	1.0	-	-	-	-
IFN-γ (20 ng/mL)	0.1	155.4	151.1	1.0	≈30↓	≈16↑	≈15↑	≈1
UVB (80 mJ/cm^2^)	0.6	13.3	14.6	0.9	≈5↓	≈1	≈1	≈1

## Data Availability

The raw data supporting the conclusions of this article will be made available by the authors upon request.

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
