# Peer review of "Exploring the Surface: Sampling of Potential Skin Cancer Biomarkers Kynurenine and Tryptophan, Studied on 3D Melanocyte and Melanoma Models"

_biomolecules, 2024, doi:10.3390/biom14070815_

Round 1
Reviewer 1 Report
Comments and Suggestions for Authors
The study of Hasterok et al. addresses the important topic of identifying melanoma biomarkers using novel non-invasive 3D in vitro skin models. It fits well with the scope of the Biomolecules journal. Early detection of melanoma could improve patients' survival and Authors propose that low-molecular-weight (LMW) biomarkers as highly promising molecules that can cross the skin barrier and can be samples in a non-invasive way. In their study, the Authors focused on the measurements of the Trp/Kyn ratio in the context of different stimuli (IFNg and UVB). The study has potential to provide important insights into the role of tryptophan and kynurenine secretion in skin models and provide an interesting tool for screening for novel LMW in melanoma detection or other skin diseases. There are however aspects in the study that should be improved or clarified.
Here are my more detailed comments:
1) Line 2-3 - In the title the word Biomarkers is separated incorrectly, it should be Bio-markers.
2) Line 27 – The statement that UVB led to changes in IDO-1, Trp and Kyn levels were changed is an overstatement that put in the abstract can lead to readers to draw false conclusions. Only Kyn levels in mm model were significantly changed.
3) Line 37 – Probably the Authors meant ‘this approach’ instead of ‘the approach’
4) Line 118 – To which diagnostics the Authors are referring to? It is not clear from the text
5) Line 131 – In this part of Introduction the Authors claim that IDO-1 overexpression and alterations of Trp/Kyn ratio are observed across all stages of the disease. However they failed to observe those differences in their model (IHC IDO-1 stainings of mm and mc models in controls, the basolateral and topical concentrations of Trp and Kyn, the ratios of Trp/Kyn in controls). I would like to ask the Authors for explanations why they think this is the case?
6) Line 171 – In the whole paper there is no explanation about how basolateral and topical sampling looks like. This is necessary to be described at least in the methods section, and advisable to be included also in the results part where the data from both samplings are discussed. It would be easier to follow the paper if the scheme of 3D skin model procedure (which seems to be quite sophisticated, at least in terms of the time it takes for the final differentiation of the model) and types of sampling and treatements are presented in the form of scheme in one of the Figures.
7) Line 234 – Why tonsils were used for the IDO-1 positive control?
8) Line 359 – In the RT-qPCR data, from which cells the mRNA was extracted? Was it the whole model dissociated or cells were dissociated and melanocytes/melanoma cells were sorted out?
9) What is known about the source of the Kyn, Trp and IDO-1 in the skin? Which cell types secrete Kyn and Trp and which ones express IDO-1?
10) Section 3.2. – I have a main remark here. Reading the details of the 3D model system procedure it looks like Authors seeded the same numbers of melanocytes and melanoma cells and then the cells were cultured for at least next 18 days. In general, melanocytes and melanoma cells differ in the proliferation rates, with melanoma cells displaying higher proliferation rates. If the system was cultured for so long there are big cell number differences at the start of the stimulations and sampling phase. What is known about the secretion (or intake) of the Kyr and Trp in terms of differences between both cell types (melanoma and melanocytes)? There is a chance that all the observed effects are the result of the cell numbers differences between two models (mm and mc) and are not model-specific. The claim that there is an effect of IFNg and UVB radiation on melanocytes and melanoma proliferation cannot be make based on the provided IHC data (Fig. 3 and 4). To conclude on proliferation rate the Authors need to provide the quantification. Providing one representative pictures that contain sometimes 1 or 2 melanocytes is not enough to draw conclusions on cells proliferation rate in this model. Please provide the quantification of the proliferation of melanocytes and melanoma cells and explanation of why there are only 1-2 melanocytes in the field of view on figure 3e and 3h. Did the melanocytes survived the whole differentiation process? What is the viability of the cells in the mm and mc model?
11) Line 390 – Please mark the cells that are described (arrows, triangles etc) so that it is easier to understand what the reader should pay attention to.
12) Line 399 – as the tissue sections used for the IDO-1 staining and Melan-A are not the same sections, and no double staining was performed for IDO-1 and Melan-1, it is not possible to distinguish between those two cell types and the conclusions should be toned down or removed.
13) Line 410-411 – what can be an explanation? Was it quantified or once again it is based on the IHC and H&E stainings? The Authors should provide more pictures of all the stainings from Figure 3 and 4 in the supplementary Figures that will support their claims.
14) Line 419-420 – The picture shows one melanoma cells. The conclusion that UVB caused reduction of proliferation is an overstatement and this conclusion cannot be based on this one picture provided. Maybe the cells died under UVB, did the Authors checked that scenario? More pictures should be provided as mentioned above.
15) Line 445-447 – This is not reflected in the data and an example of another overstatement. There is no different present and not even trends. If the Authors see some trends based on the p values please provide the values.
16) Line 464 – Higher than what?
17) Line 466 - If anything - the Kyn increased concentration is more pronounced than Trp decrease which is not at all visible in the data.
18) Line 483-484 – This is an overstatement, in the Figure 7b there are no differences observable. The Authors need to tone down all the conclusions on the non-significant data and if trends were seen they need to provide the p values.
19) Does the media used in the study contain the measured amino acids: Kyn, Trp, Tyr, Phe? If yes, how the Authors controlled for the effect of those on the generated data?
20) Line 495 – which biological factors? Please explain
21) Line 502 - the significance is not marked for control vs IFNg in both models. Is it significant?
22) Are tables 2 and 3 representing the same data as shown in Fig. 8?
23) Line 595 – as mentioned above, the conclusion about the proliferation needs to be supported by quantification.
24) Line 600-602 - Melanocytes also have melanin. Also - the Authors suggest here that melanin shields melanoma cells from the harmful effects of UVB referring to the UVB-dependent decrease of cells (questionable from the start).
25) Line 636-637 – this sounds like the same thing. Please rephrase.
26) Line 664-665 -phenylalanine and tyrosine are substrates for melanogenesis and can be affected by the differentiation state of melanoma cells. Can Authors elaborate on those effects? How safe it is to use those two amino acids as the controls?
27) As discussed above conclusions need to be toned down and be supported by the data shown.
28) Line 741 – The aspect of sampling needs to be better described at the beginning of the paper.
Reviewer 2 Report
Comments and Suggestions for Authors
While the study is well designed, the paper would greatly benefit from some additions in the conclusions and future direction part of the Discussion (section 4).
The 3D models were created using a single fibroblast cell line, a single melanoma cell line and a not particularly well characterized primary keratinocytes, all from a commercial source, but not from the same individual or patient.
Skin exhibits an astounding degree of variation across and between different racial, ethnic or smaller population groups not only regarding the melanin expression and distribution, but also regarding the various, genetically determined metabolic and immune activities of its various cells. The authors mention the potential effect of melanin in the melanoma cells. It would be interesting to know how variable are the IDO-1 activity, IFN-gamma expression, Trp and Kyn levels and ratios in various individuals from different ethnic group or skin types. Are there any significant differences between individuals who tend to tan well vs. those who rather burn but do not tan well (e.g. redheads)?
It is well known that even primary melanomas exhibit great biological variability, since melanoma can arise from different driving mutations (e.g. BRAF vs NRAS mutations) leading to distinct biology which may affect metabolic pathways too. Therefore, it would be prudent to plan future experiments using 3D models with different melanoma cell lines, fibroblasts and even keratinocytes, and also ones constituted from primary cells coming from the same individual or patient.
I think these experiments could clear the way for establishing if the proposed markers could be truly used as a widely usable true biomarkers for early detection of primary melanoma.
Also, it would be interesting to see in some future experiments what would be the combined effect of treating the 3D models with both INF-gamma and UVB, since it would be mimicking a realistic situation where a patient developing melanoma had a very recent UVB exposure.
I do not suggest any further experiments to be performed for the current paper, just a slight expansion of the future directions section.
Two small remarks:
Line 246:
I think the authors should reach out to Qiagen's technical service and clarify the exact role of the 70% ethanol addition step during the RNA purification.
Line 396:
Did the authors mean "little to no increase" instead of reduction?
Comments on the Quality of English LanguageSlight editing could be helpful, but overall the English language quality is very good.
Round 2
Reviewer 1 Report
Comments and Suggestions for Authors
I want to thank the Authors for providing the corrections and extensive explanations for my comments. I am impressed by the amount of work the Authors put into their cover letter and their willingness to tone down many overstatements. In the present form, the manuscript provides a decent description of the obtained results and can serve as a foundation for further studies under the Trp/Kyn pathways in early non-invasive skin detection. In my opinion, the manuscript can be published after minor corrections:
1. In Figures 3 and 4 I do not see any labels a-i and any marks with E and D of the dermis and epidermis part, I see only a black frame which I believe is a matter of some problem with the picture conversion. Please make sure this is corrected before the paper is published.
2. The title of section 3.2 should also be changed accordingly and the Authors should remove the proliferation aspect.
I would also like to note that unfortunately, it was not easy to follow the changes in the main part of the manuscript as the changes were not marked in the PDF file, and the description of changes in the cover letter where the Authors provided the line numbers were not matching the lines in the provided PDF.
